# Hepatitis B virus core protein allosteric modulators can distort and disrupt intact capsids

Christopher John Schlicksup[1], Joseph Che-Yen Wang[1,2], Samson Francis[3], Balasubramanian Venkatakrishnan[1], William W Turner[3], Michael VanNieuwenhze[4], Adam Zlotnick[1]*

[1]Department of Molecular and Cellular Biochemistry, Indiana University, Bloomington, United States; [2]Indiana University Electron Microscopy Center, Bloomington, United States; [3]Assembly Biosciences, Carmel, United States; [4]Department of Chemistry, Indiana University, Bloomington, United States

**Abstract** Defining mechanisms of direct-acting antivirals facilitates drug development and our understanding of virus function. Heteroaryldihydropyrimidines (HAPs) inappropriately activate assembly of hepatitis B virus (HBV) core protein (Cp), suppressing formation of virions. We examined a fluorophore-labeled HAP, HAP-TAMRA. HAP-TAMRA induced Cp assembly and also bound pre-assembled capsids. Kinetic and spectroscopic studies imply that HAP-binding sites are usually not available but are bound cooperatively. Using cryo-EM, we observed that HAP-TAMRA asymmetrically deformed capsids, creating a heterogeneous array of sharp angles, flat regions, and outright breaks. To achieve high resolution reconstruction (<4 Å), we introduced a disulfide crosslink that rescued particle symmetry. We deduced that HAP-TAMRA caused quasi-sixfold vertices to become flatter and fivefold more angular. This transition led to asymmetric faceting. That a disordered crosslink could rescue symmetry implies that capsids have tensegrity properties. Capsid distortion and disruption is a new mechanism by which molecules like the HAPs can block HBV infection.

DOI: https://doi.org/10.7554/eLife.31473.001

*For correspondence:
azlotnic@indiana.edu

## Introduction

Worldwide, an estimated 240 million people suffer from chronic hepatitis B virus (HBV); infection is most often acquired in early childhood. Over time, chronic HBV can lead to liver disease including liver failure, cirrhosis, and hepatocellular carcinoma, contributing to more than 700,000 deaths each year (*Gish et al., 2015*). A vaccine directed against surface protein is preventative but not therapeutic. Entry inhibitors prevent new infection but are unlikely to affect maintenance of a chronic infection (*Volz et al., 2013*). Reverse transcriptase inhibitors directed against the viral polymerase suppress viremia and improve liver health, but even after years of treatment are rarely curative (*Gish et al., 2007*; *Shi et al., 2015*). Interferon α and derivatives can stimulate HBV clearance in a small subset of patients (*Gish et al., 2015*). The viral core protein (Cp) has become an important target for developing direct-acting antivirals (DAAs) (*Zlotnick et al., 2015*).

The HBV Cp is present at virtually every step of the virus lifecycle in an infected cell (*Zlotnick et al., 2015*). In the nucleus, Cp is associated with viral nucleic acid (*Bock et al., 2001*; *Li et al., 2010*). In cytoplasm, Cp assembly packages viral polymerase complexed with an RNA copy of the viral genome and has an active role in reverse transcription of the relaxed circular mature genome (*Tan et al., 2015*). Cp also has signals that direct core intracellular trafficking and secretion of enveloped cores (*Kann et al., 2007*). When transported to the nucleus, for new infection and to

**eLife digest** Viruses are simple structures formed of genetic information wrapped inside a shell. For the hepatitis B virus, this casing looks like a soccer ball. It is composed of 240 copies of the same protein, arranged in a pattern of pentagons and hexagons. These proteins form a protective shield for the virus' genetic information: they also interact with the cells of the host during key events of the virus' life cycle.

When the hepatitis B virus infects a cell, it hijacks the cellular machinery to replicate. New shell proteins are produced and assemble within the cell. A type of potential antiviral drug called a CpAM disrupts this process: it causes the shell to assemble too early and inaccurately, which impairs the life cycle of the virus. However, a CpAM can bind to the shell even after it has already assembled. How this binding affects the virus is still unclear.

Here, Schlicksup et al. attach a fluorescent molecule to a CpAM, and use a cutting-edge microscopy method to look at the structures at the atomic level. This makes it possible to examine in detail how the CpAM attaches to a correctly formed virus shell. Schlicksup et al. show that when the CpAM binds to the shell, it disrupts and sometimes even breaks the soccer-like pattern of the shell: the hexagons flatten, and the pentagons buckle.

These misshaped shells could prevent the virus from interacting with the cellular structures necessary for infection or prevent it from releasing the virus' genetic information. This is a new antiviral mechanism for a CpAM. By acting both before and after the shell has assembled, the CpAM targets the virus at different points of its life cycle.

Hepatitis B affects over 240 million people worldwide. While a vaccine exists, there is still no cure for it. A better understanding of the physics of the virus' shell and the mode of action of CpAMs could lead to better drugs against the disease.

DOI: https://doi.org/10.7554/eLife.31473.002

maintain infection, the core disassembles to release its contents. Curiously, approximately 90% of the capsids from enveloped particles are empty and the rest contain the viral genome (*Ning et al., 2011*). This leads to the hypothesis that assembly can be nucleated by a polymerase-viral RNA complex for virions or spontaneously by Cp. This bifurcation of paths suggests that assembly is susceptible to disruption. The oligomeric nature of Cp function makes it a particularly attractive antiviral target because even when DAA-resistant mutants arise, residual wildtype protein co-assembled into a complex retains DAA sensitivity (*Tanner et al., 2014*).

The HBV Cp is a 183-residue, homodimeric protein (*Venkatakrishnan and Zlotnick, 2016*). The first 149 residues form the assembly domain. Each monomer contributes two helices to a four-helix bundle that forms the intradimer interface (*Venkatakrishnan and Zlotnick, 2016*). The final 34 residues of the full length Cp form the disordered nucleic acid-binding C-terminal domain. This domain is optional for capsid assembly but required for nucleic acid packaging; this region also carries putative nuclear localization and nuclear export signals (*Li et al., 2010*; *Kann et al., 2007*). Cp assembles into icosahedral 90-dimer T = 3 capsids and 120-dimer T = 4 capsids. In capsids, dimers are arranged in a pattern of fivefold vertices and quasi-sixfold vertices. Thus, owing to quasi-equivalence, a T = 3 capsid is comprised of 60 AB dimers and 30 CC dimers (with true twofold symmetry) and a T = 4 capsid is comprised of 60 AB and 60 CD dimers. In vitro assembly of the assembly domain or full-length Cp yields about 90% T = 4 capsids; a similar ratio is observed in vivo (*Venkatakrishnan and Zlotnick, 2016*; *Stannard and Hodgkiss, 1979*).

Studies of in vitro assembly of Cp have provided insights into the biology of HBV, the general problem of capsid assembly and the mechanism of assembly-directed antivirals. It is hypothesized that assembly is initiated by a conformational transition of Cp to an active state, suggesting a basis for regulated assembly in vivo (*Packianathan et al., 2010*). The net association energy is weak, but since each of the many Cp dimers forming a capsid are tetravalent, the result is a globally stable structure (*Zlotnick, 2003*). In vitro assembly of HBV Cp is sensitive to ions and temperature, supporting the hypothesis that assembly can be allosterically regulated (*Stray et al., 2004*; *Ceres et al., 2004*). Furthermore, there is a close relationship between the familiar phenomenon of allostery and the structural principle of tensegrity in virus capsids (*Domitrovic et al., 2013a*).

Putative Cp allosteric effectors (e.g. heteraryldihydropyrimidines or HAPs) have been discovered that have antiviral effect (*Deres et al., 2003*; *Stray et al., 2005*; *Stray and Zlotnick, 2006a*). These small molecules ostensibly act by inducing an assembly-active state and by stabilizing Cp-Cp interactions by filling a pocket at the interdimer interface between dimers of the quasi-sixfold (*Venkatakrishnan and Zlotnick, 2016*; *Bourne et al., 2006*). We refer to this class of molecules, which now includes a diverse set of chemistries (*Pei et al., 2017*), as Core protein Allosteric Modulators (CpAMs). The well-characterized mechanism for CpAM action is to over-initiate assembly, forming empty capsids and large aberrant structures. Both products act to sequester Cp from its biological functions. We suggest that CpAMs likely have additional effects on the HBV lifecycle and are developing fluorescent CpAMs as tools to better probe their effects both in vitro and in cells.

Here, we used cryo-electron microscopy (cryo-EM) as a means to look at capsids in complex with the fluorescent CpAM HAP-TAMRA. The fluorescent CpAM provides a useful signal for binding, and we hoped that the extra mass would assist identifying the molecule in structural studies. Cryo-EM provides advantages compared to previous studies using crystalized forms of Cp, in that capsid-wide deformations can be observed, including asymmetric deformations. This establishes a basis for further antiviral development focused on disruption of capsid structure that is distinct from the previously established mechanisms of CpAM activity.

## Results

### HAP-TAMRA is a probe of core protein and CpAM interaction

To better understand the relationship between CpAMs and core protein, we synthesized a fluorescent CpAM, HAP-TAMRA (*Figure 1*). HAP-TAMRA's HAP13 core (*Bourne et al., 2008*) was extended with a three-carbon linker appended with a tetramethylrhodamine (TAMRA) dye. As the HAP13 core used in this synthesis was a racemic mixture, only half the compound is expected to be active (*Deres et al., 2003*). We anticipated the TAMRA moiety to extend into the capsid lumen from the HAP-binding pocket and cause minimal disruption to interactions between the HAP and protein. To verify the activity of HAP-TAMRA, we examined its effect on purified dimeric assembly domain, Cp149. We observed that assembly reactions of purified core protein in the presence of HAP-TAMRA produced large aberrant products that are characteristic of HAP-induced assembly (*Figure 2b*). For reference, assembly reactions without CpAMs (*Figure 2b*) led to particles morphologically consistent with an icosahedral shell.

In this work, our focus was the binding of HAP-TAMRA to pre-assembled capsids, similar to those in *Figure 2a*. We first evaluated formation of the capsid HAP-TAMRA complex using size exclusion chromatography with a diode array absorbance detector (*Figure 2c,d*). Capsids co-eluted with bound HAP-TAMRA in the capsid fraction. Free HAP-TAMRA eluted at the end of the column volume, as expected for a small molecule. TAMRA absorbance spectra associated with the capsid and free probe were distinctly different. The free probe exhibited the familiar absorbance profile of TAMRA, with a major peak near 555 nm, and a shoulder at 520 nm. The absorbance spectrum for capsid-bound HAP-TAMRA was shifted, with a major peak near 520 nm and a shoulder around 550 nm that resembled examples of spectra from π-stacked TAMRA complexes (*Adachi et al., 2014*). Using the change in signal at 520 nm, we monitored the amount of capsid- bound HAP-TAMRA. We also examined the utility of the ratio of 520/555 nm absorbance as a measure of binding (*Figure 2d*). For both the absorbance at 520 nm and the 520/555 ratio, the signal associated with capsid peaked at about one HAP-TAMRA per dimer. These data also suggest that HAP-TAMRA binds capsid with high affinity compared to the 8 μM Cp dimer used in these experiments. The appearance of the 520 nm peak under these conditions was restricted to the capsid fraction as shown by the absence of a ratiometric change in the free fraction (*Figure 2d*).

This absorbance signal change is also visible in samples that are not purified by size exclusion (*Figure 3a*). We chose to use the increase in absorbance at 520 nm to study the kinetics of binding, (*Figure 3b–f*). Using 2.5 to 40 μM HAP-TAMRA, we observed that binding was surprisingly slow, reaching half-saturation in approximately 10 min at all concentrations. When the different kinetic trajectories were normalized to the same maximum and minimum values, zero and one, it was evident that they were identical (*Figure 3c*), indicating that the rate of binding under these conditions was independent of HAP-TAMRA concentration. Therefore, we examined binding kinetics in terms of a

**Figure 1.** Synthesis of HAP-TAMRA. HAP-TAMRA is a derivative of HAP13 (*Bourne et al., 2008*) with a tetramethylrodamine (TAMRA) moiety.
DOI: https://doi.org/10.7554/eLife.31473.003

first order reaction. Slow kinetics can indicate that the binding pocket is not usually accessible to the ligand, but that a conformational transition of the protein allows the pocket to open transiently. Curve fits using a single exponential did not fit the observed binding kinetics but an association model with two phases of equal magnitude matched the kinetic trajectories. The average for the fast half-life was 0.6 ± 0.1 min and for the slow half-life was 7.1 ± 1.4 min (*Figure 3b*). This HAP-TAMRA concentration independence and the slow binding rate strongly suggest that the primary barrier for HAP binding is a conformational transition at the site of the binding pocket. Because the signal for binding is an absorbance shift consistent with stacking of two or more TAMRA moieties, examination of the absorbance when the reaction was complete should also give an indication of the mechanism of binding. We observed that the change in absorbance increased linearly until capsids were saturated at one HAP-TAMRA per dimer (*Figure 2d*). The change in absorbance per bound HAP-TAMRA was also evaluated (i.e. the change in extinction coefficient) and found to be 11200 ± 4% $M^{-1}$

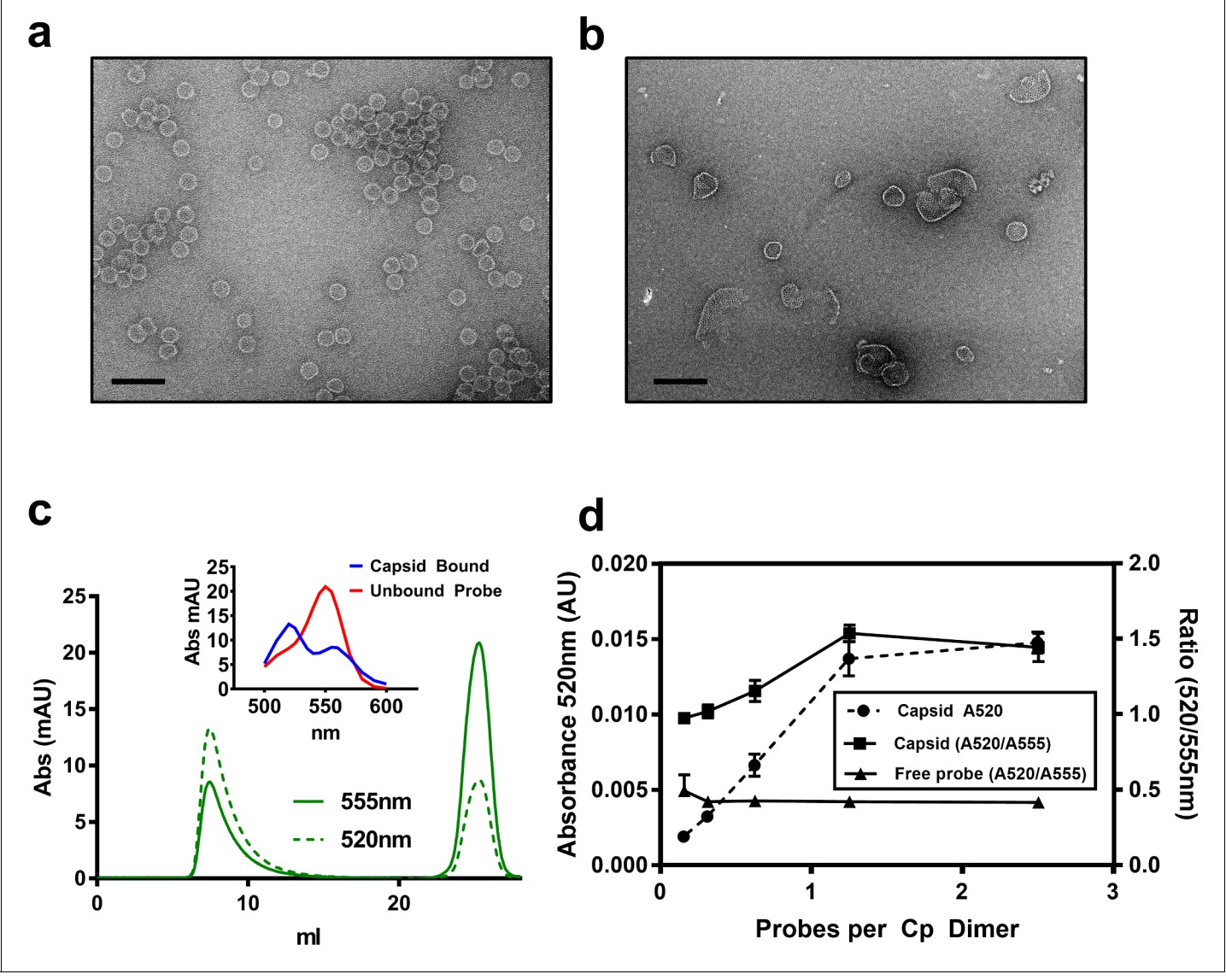

**Figure 2.** HAP-TAMRA drives core protein assembly. (a) A negative stain micrograph of a typical assembly reaction using purified core protein. (b) Like other HAPs, stoichiometric excess of HAP-TAMRA drives assembly of non-capsid polymers of core protein. (c) A representative size exclusion chromatograph, monitored at 555 nm to pick up TAMRA, shows that HAP-TAMRA co-elutes with capsid. (inset) Spectra of the capsid-bound HAP-TAMRA (blue) and free forms show an unusual chromatic shift, suggestive of $\pi$-stacking of TAMRA moieties in the capsid. (d) Using the absorbance shift as an indicator (either the value at 520 nm or the ratio of 520/555 nm), we find that preformed capsids saturate at a stoichiometry of about one small molecule per dimer. The x axis represents 'effective' probes per Cp dimer, considering that half the input probe is an inactive enantiomer. Each point is an average of three measurements with error bars showing the standard deviation. The scale bars in (a) and (b) are 100 nm.

DOI: https://doi.org/10.7554/eLife.31473.004

(*Figure 3e*). This result indicates that there is the same degree of TAMRA–TAMRA interaction in saturated capsids and in capsids where there was on average only one HAP-TAMRA per capsid. The change in absorbance spectrum suggested TAMRA-TAMRA interaction that would be expected to cause static quenching; this prediction was confirmed in *Figure 3f* where the residual fluorescence is largely due to unbound probe.

## HAP-TAMRA distorts capsids

Previous structural studies of HAPs with HBV core protein have depended on crystallography, which selects for regular complexes. Thus, when we looked at cryo-EM 2D class averages of HAP-TAMRA-

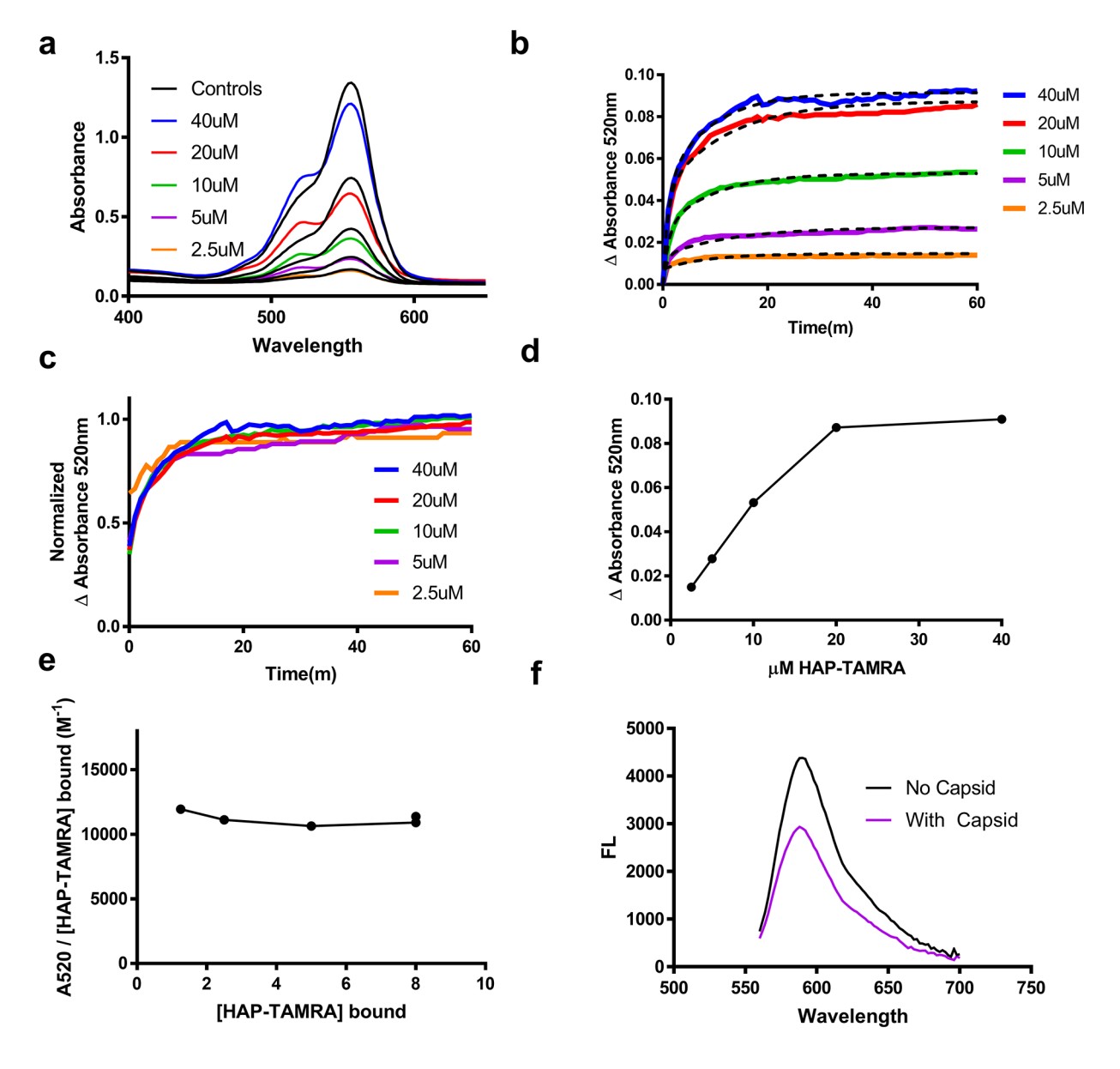

**Figure 3.** HAP-TAMRA absorbance and fluorescence change in response to binding capsid. (a) HAP-TAMRA in solution for a matched pair of samples with 8 µM capsid (colored lines) and without (black lines). In the presence of capsid, the absorbance spectrum shifts towards one that resembles a π-stacked TAMRA dimer (*Adachi et al., 2014*). The change is not as dramatic as in *Figure 2c*, due to the presence of free dye in sample that have not been purified by SEC. (b) The change in absorbance at 520 nm can be used to observe kinetics of binding different HAP-TAMRA concentrations to 8 µM capsid, showing the time dependence of forming the samples in panel a. The kinetic traces are averages of three independent experiments. The dashed lines fit to a two-phase exponential association. The half-life for binding is between 5 and 10 min. (c) Further normalization of the kinetic curves by their maximum value gives a visual expression of the similarity of the half-lives, and reiterates that the kinetics appear to be similar across input concentrations (d) The time course in b was monitored for 350 min. Shown are the endpoint values of the change in 520 nm absorbance. The result is comparable to the one in *Figure 2d*, where the bound probe is isolated from the free probe. (e) Assuming the maximum amount of HAP-TAMRA binds to the protein in each condition, we can calculate the concentration of HAP-TAMRA bound. Plotting extinction coefficient for the ΔA520nm signal is constant across the titration, with a value of 11196 $M^{-1}$ ± 4%. (f) A demonstration that fluorescence is also quenched upon binding using 40 µM HAP-TAMRA, 8 µM Cp dimer. In the presence of capsid, we estimate that 75% of the HAP-TAMRA was unbound, accounting for most of the residual fluorescence.

DOI: https://doi.org/10.7554/eLife.31473.005

saturated capsids we anticipated a field of isometric particles similar to apo-capsids (drug free) (*Figures 2a* and *4b*). However, in both raw micrographs and 2D class averages, the heterogeneity of particle shapes with bound HAP was evident (*Figure 4a,c*): many particles had sharp corners, facets, and irregular outlines. To distinguish 'distorted' from the normal capsids, we performed 2-D class averaging to compare particles from Cp149-apo (*Figure 4b*) and Cp149+HAP TAMRA (*Figure 4c*) datasets. In this technique, particles are grouped based on dominant features of the microscopic image, rotationally oriented, and presented as an average. Apo-capsids (*Figure 4b*) had a round cross section with obvious spikes and distinct internal features. The capsids with HAP-TAMRA had a broad distribution of asymmetric shapes with ellipsoidal character and distinct faceting.

Because of the structural irregularity of these capsids, they would be of limited use in pursuit of a high-resolution reconstruction. Therefore, we examined Cp150 capsids; Cp150 is a variant of Cp149 with an engineered disulfide bond linking the disordered C-terminal tails when in capsid form (*Lee et al., 2017*). Our rationale was that the inter-dimer disulfides would provide a soft positional restraint on the capsid, and prevent it from adopting conformations with extreme asymmetry due to bound probe. The 2D class averages of Cp150 capsids incubated with HAP-TAMRA (*Figure 4c*) showed regular and approximately spherical particles; suitable for further detailed structural analysis.

## Cryo-EM structures of T = 4 and T = 3 capsids bound to HAP-TAMRA

To examine HAP-bound capsids, we reconstructed T = 4 and T = 3 Cp150-HAP-TAMRA capsids to 4.0 Å and 3.7 Å resolution, respectively, from a single dataset (*Figures 5* and *6*; *Table 1*). The resolution was sufficient to position and rebuild molecular models of protein and HAP-TAMRA (*Video 1*). These structures are, to our knowledge, the first instance of using cryo-EM to visualize a drug bound

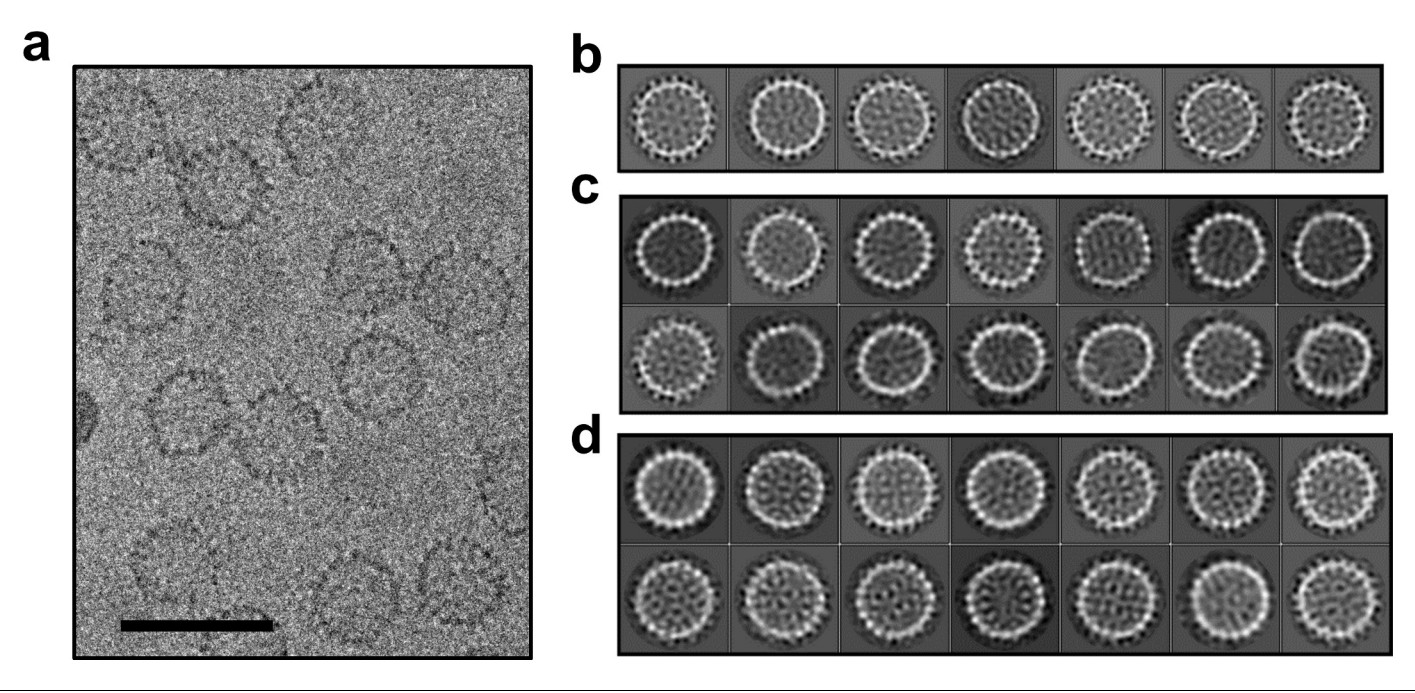

**Figure 4.** Cryo-microscopy shows that HAP-TAMRA distorts capsids. (**a**) A cryo-micrograph shows a section of a field of Cp149 capsids treated with HAP-TAMRA. Particles appear asymmetric and have marked angles. None have the spherical cross-section typical of Cp149 capsids. (**b–d**) Cryo-micrographs of capsids were subject to 2D class averaging to identify their major characteristics. In each case, the classes are sorted by descending class distribution, (**b**) For apo-capsids of Cp149, lacking any HAP-TAMRA, all classes have a circular cross-section and a periphery of spikes. These seven classes encompass more than 80% of the Cp149 capsid particles sampled. (**c**) For micrographs of Cp149 particles soaked with HAP-TAMRA the most common class has an elliptical cross-section. Other classes have distinct and asymmetric faceting. The data in panels a and c are independent acquisitions of the same experimental conditions, demonstrating the repeatability of the effect. (**d**) The fourteen most populated classes from a micrograph of cysteine cross-linked Cp150 capsids treated with HAP-TAMRA resemble those of apo-Cp149 in panel b.
DOI: https://doi.org/10.7554/eLife.31473.006

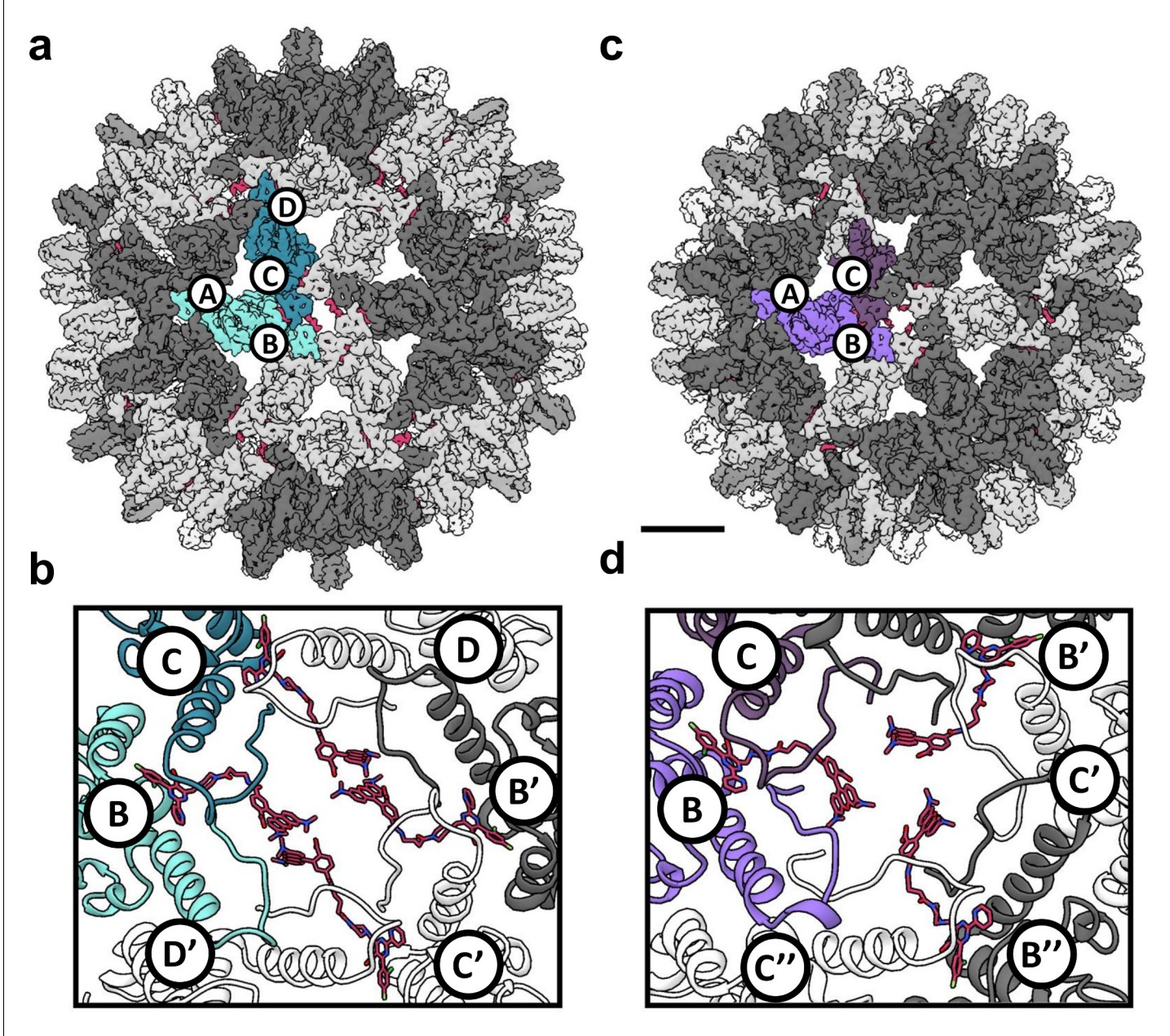

**Figure 5.** Reconstruction of T = 4 and T = 3 particles. (a) A T = 4 capsid. A single asymmetric unit is composed of two dimers, and by convention the component subunit chains are labeled with letters A,B and C,D. The two subunits that form the AB dimer (light green) and a CD dimer (dark green) are highlighted. The HAP-TAMRA molecules are highlighted (red), and bind around a quasi-sixfold (icosahedral twofold) symmetry axis. The remaining AB dimers are shown in darker grey than the CD dimers. The scale bar is 5 nm and also applies to panel (c). (b) A single quasi-sixfold cluster from a T = 4 capsid. The coloring scheme is that same as in (a) except that all HAP-TAMRA molecules at the quasi-sixfold are highlighted. (c) A T = 3 capsid with the asymmetric unit comprised of an AB dimer (light purple) and one half of a CC dimer (dark purple) highlighted. (d) As per (b) except a T = 3 quasi-sixfold has threefold symmetry and three bound HAP-TAMRA molecules. In panels (b) and (d) symmetry-related subunits are denoted by the prime or double prime symbols.

DOI: https://doi.org/10.7554/eLife.31473.007

to a virus capsid and to show that T = 3 capsids bind CpAMs (*Video 2*). Cryo-EM also allowed high resolution analysis of changes to quaternary structure without the potential bias of crystallographic packing constraints. The T = 4 capsids were slightly larger than crystallographic structures, though each crystal structure has shown unique features (*Venkatakrishnan and Zlotnick, 2016*). The

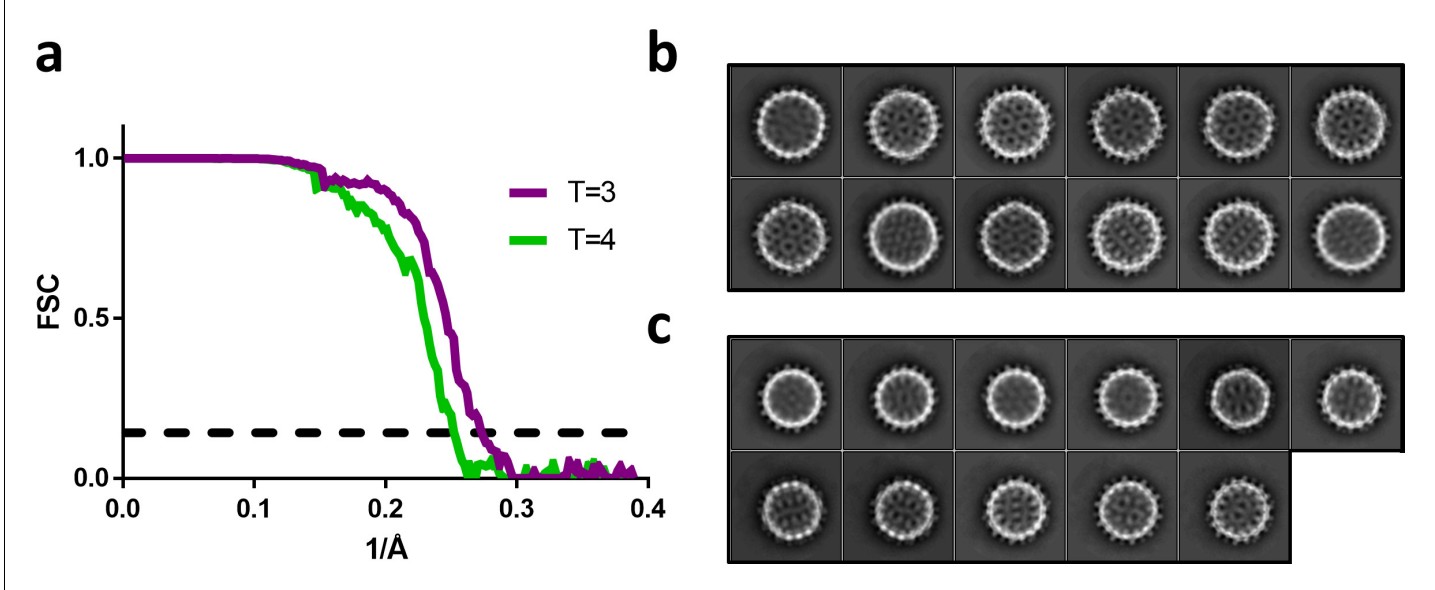

**Figure 6.** Fourier shell correlation and 2D class averages of T = 4 and T = 3 particles. (a) Fourier shell correlation for the T = 4 and T = 3 reconstruction in green and purple, respectively. The dashed line indicates a correlation of 0.143. (b) The most populated 2D class averages from the T = 4 dataset reconstructed in *Figure 5a*. (c) The most populated 2D class averages from the T = 3 dataset reconstructed in *Figure 5c*.

DOI: https://doi.org/10.7554/eLife.31473.008

comparison to the T = 3 capsid, in particular, shows commonalities and differences in Cp–Cp interactions.

Besides the difference in diameter and the differing number of component Cp dimers, the local difference in T = 4 and T = 3 architecture is that quasi-sixfold vertices have twofold and threefold symmetry, respectively. A T = 4 quasi-sixfold has two repeats (B, C, D, B', C', and D') subunits and in T = 3 there are three repeats (B, C, B', C', B'', and C'') (*Figure 5*). In T = 4 capsids, quasi-sixfold hexamers have four HAP-TAMRA molecules, while the T = 3 quasi-sixfold hexamers accommodate three HAP-TAMRA molecules. The HAP13 portion of the probe is nestled in the pocket between two adjacent dimers with the linker extending from the pocket, positioning the TAMRA near the center of the quasi-sixfold pore. In the T = 4 quasi-sixfold, the HAPs are in B, C, B', and C' pockets, which are respectively capped by C, D, C' and D' subunits (*Figure 5b*). In the T = 3 particles, HAPs are in B pockets capped by C subunits (*Figure 5d*). These results place the HAP moiety of HAP-TAMRA in the same pocket associated with other CpAMs HAP1, AT130, HAP18, and sulfamoyl benzamides. (*Bourne et al., 2006*; *Venkatakrishnan et al., 2016*; *Zhou et al., 2017*) No small molecule density was observed at the corresponding A (fivefold) and D sites, which are occluded by A and B subunits, respectively. The pocket of the T = 3 C subunit is occluded by the neighboring B subunit, resembling the interaction of the T = 4 D subunit with its neighboring B subunit.

Despite the quaternary differences between T = 3 and T = 4 capsids, the local environment of the HAP-TAMRA in the HAP pocket was remarkably similar (*Figure 7*). The HAP13 had extremely well-ordered density in the pocket as do most protein side chains in the region (visualized in *Figure 7e* at a contour of 3.6 σ, where σ is the number of standard deviations from the mean). The quality of HAP-TAMRA density gradually decays in the TAMRA region, which is visible as a loss of density in *Figure 7*, and was also seen as a decay in local resolution in *Figure 8*. This observation is consistent with fewer positional constraints in the exposed TAMRA moiety, and higher molecular motion. In particular, the T = 4 C subunit the TAMRA density is only contiguous at 2.6 σ. For the T = 4C subunit, the loss of linker density makes precise fitting the connection from the HAP to the TAMRA ambiguous. However, linker density clearly points towards a planar mass that is parallel to and 'stacked' on the TAMRA from the B' subunit. By symmetry, there is an equivalent interaction between TAMRAs from C' and B. This resulting best fit into the available density should not be considered an exclusive model. However, as 80–90% of the capsids have T = 4 symmetry, we suggest

**Table 1.** Image reconstruction statistics.
The table summarizes the statistics for data collection, structure determination, and refinement.

| Collection/Refinement Parameters | T=3 | T=4 |
| --- | --- | --- |
| **Data Collection** | | |
| Microscope | FEI Titan Krios | FEI Titan Krios |
| Voltage (kV) | 300 | 300 |
| **Dose (e⁻ / Å²)** | 33 | 33 |
| Detector | Gatan K2 Summit | Gatan K2 Summit |
| Pixel size (Å) | 1.285 | 1.285 |
| Defocus range (µm) | 0.5-3.5 | 0.5-3.5 |
| **Reconstruction (RELION)** | | |
| Micrographs | 679 | 679 |
| Particle number (Initial) | 15,066 | 24,823 |
| Particle number (Final) | 13,746 | 16,008 |
| Symmetry | Icosahedral | Icosahedral |
| Box size (pixels) | 380 | 380 |
| Accuracy of rotations | 0.434° | 0.432° |
| Accuracy of translations (pixels) | 0.50 | 0.60 |
| **Sharpening B-factor (Å²)** | -140.93 | -182.60 |
| Final resolution (Å) | 3.67 | 3.97 |
| EMDB accession code | 7295 | 7294 |
| **Model Refinement (PHENIX)** | | |
| Cross correlation (Whole Volume) | 0.764 | 0.776 |
| Cross correlation (Masked) | 0.844 | 0.819 |
| **Ramachandran Plot** | | |
| Outliers | 0.00% | 0.00% |
| Allowed | 4.3% | 7.8% |
| Favored | 95.7% | 92.2% |
| **PDB accession code** | **6BVN** | **6BVF** |

DOI: https://doi.org/10.7554/eLife.31473.009

that these TAMRA-TAMRA interactions are the cause of the observed change in the optical signal (*Figure 2c,d*).

## HAPs promote increased faceting of the capsid

When comparing T = 3 to T = 4, and T = 4 CpAM-bound to T = 4 apo structures, we found that local similarities in protein structure do not translate to similarities one or two dimers away. Long range differences in quaternary structure were apparent when overlaying entire hexamers, for example overlaying T = 3 and T = 4 quasi-sixfold hexamers based on one set of AB dimers (*Figure 9*). The next subunit around the quasi-sixfold, the D subunit (T = 4) or C subunit (T = 3), show pronounced differences while the dimer on the far side of quasi-sixfold is completely misaligned. The dramatic effect arises because the small angular differences are amplified by distance. Another relevant example is the assembly-inactive core protein mutant Y132A.(*Packianathan et al., 2010*; *Zhou et al., 2017*; *Klumpp et al., 2015*; *Qiu et al., 2017*) The Y132A mutation disrupts normal interdimer interaction allowing crystal contacts that form hexagonal layers reminiscent of but distinct from subunit interactions in capsids. In this cryo-EM structure, density for Y132 is well-resolved (*Figure 7e*). While the Y132 mutant supports high resolution crystallography (*Packianathan et al., 2010*; *Klumpp et al., 2015*), it is impossible to predict CpAM effects to a capsid in that context. Such CpAM-induced quaternary differences between HBV capsids do not appear arise from changes in

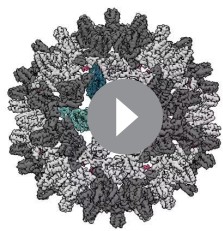

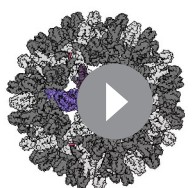

**Video 1.** A tour of a T=4 capsid (*Video 1*). Intact surface-shaded capsids, contoured at 3.6 sigma, are shown with the AB dimers surrounding fivefold vertices shaded in dark grey, quasi-sixfold are alternating dark and light gray. HAP-TAMRA is red. An asymmetric unit is colored, disrupting the color scheme of one fivefold and one quasi-sixfold. For the T=4 capsid, the AB dimer is light blue and the CD dimer is dark blue. The quasi-sixfold has two B-C-D repeats. A quasi-sixfold is then extracted from the capsid and then the grey subunits and extra HAP-TAMRAs removed from the asymmetric unit. Density and then models are removed from the C HAP-TAMRA and CD dimer. The AB dimer with the B HAP-TAMRA are trimmed to show only the HAP pocket to show that density is evident for the residues that nestle the HAP moiety. Note that the linker and part of the TAMRA show clear density even at this high contour level.

DOI: https://doi.org/10.7554/eLife.31473.015

**Video 2.** A tour of a T=3 capsid (*Video 2*) As with the T=4 move, intact surface-shaded capsids, contoured at 3.6 sigma. The AB dimers surrounding fivefold vertices shaded in dark grey, quasi-sixfold are alternating dark and light gray. HAP-TAMRA is red. An asymmetric unit is colored, disrupting the color scheme of one fivefold and one quasi-sixfold. For the T=3 capsid the AB dimer is light purple and the C subunit is dark purple. The quasi-sixfold has alternating B and C subunits. A quasi-sixfold is then extracted from the capsid and then the grey subunits and extra HAP-TAMRAs removed from the asymmetric unit. Density and then models are removed from the C HAP-TAMRA and CD dimer. In the T=3 movie this operation is repeated with the C subunit. The AB dimer with the B HAP-TAMRA are trimmed to show only the HAP pocket to show that density is evident for the residues that nestle the HAP moiety. Note that the linker and part of the TAMRA show clear density even at this high contour level.

DOI: https://doi.org/10.7554/eLife.31473.016

the binding to the HAP pocket. Rather, they arise from the way the capping subunit overlays the pocket and how these differences propagate across a capsid.

The structural differences between the HAP-bound T = 4 particle and an apo-capsid are evident when capsids are aligned based on icosahedral symmetry. It is immediately clear that the HAP-bound particle has expanded, particularly at the fivefold (*Venkatakrishnan et al., 2016*). The pairwise Cα separation between the HAP-TAMRA capsid and an apo-capsid (1QGT [*Wynne et al., 1999a*]) shows large-scale changes and regions of constraint that were not obvious when focusing on a single asymmetric unit (*Figure 10*). The cryo-EM reconstruction shows a greater expansion than seen in crystallographic structures of other CpAM-capsid complexes (*Bourne et al., 2006*; *Venkatakrishnan et al., 2016*; *Katen et al., 2013*). In comparison to the apo capsid, the displacement maximum is almost 6 Å around the 5-fold axis, and the displacement minimum is 1.7 Å around the 3-fold axis. The two-dimer asymmetric unit is rotated along the long axis of the CD dimer, near the 3-fold, leading the AB dimer to pivot upward, raising at the 5-fold. We suggest that the basis of this quaternary change is a flattening of the quasi-sixfold hexamers. Flattening quasi-sixfold and concomitantly making the 5-fold vertices protrude is likely a subdued version of what we observed for Cp149 (*Figure 4*).

An important observation is that the structural variation that leads to faceting of T = 4 capsids can also be observed by examining the cryo-EM local resolution maps (*Figure 8*). The local resolution is systematically better around the 2-fold symmetry axis, and worse around the 5-fold. This result suggests a greater positional variation of A subunits, equivalent to a crystallographic B factor. In both AB and CD dimers, the highest resolution is observed in the core of the protein, the chassis subdomain (*Packianathan et al., 2010*), and the lowest resolution, falling to about 5 Å, is at the spike tips. Protein–protein interactions around the quasi-sixfold were extremely well-ordered as were the HAP moieties of HAP-TAMRA, about 3.6 Å in the overall 4 Å resolution T = 4 structure.

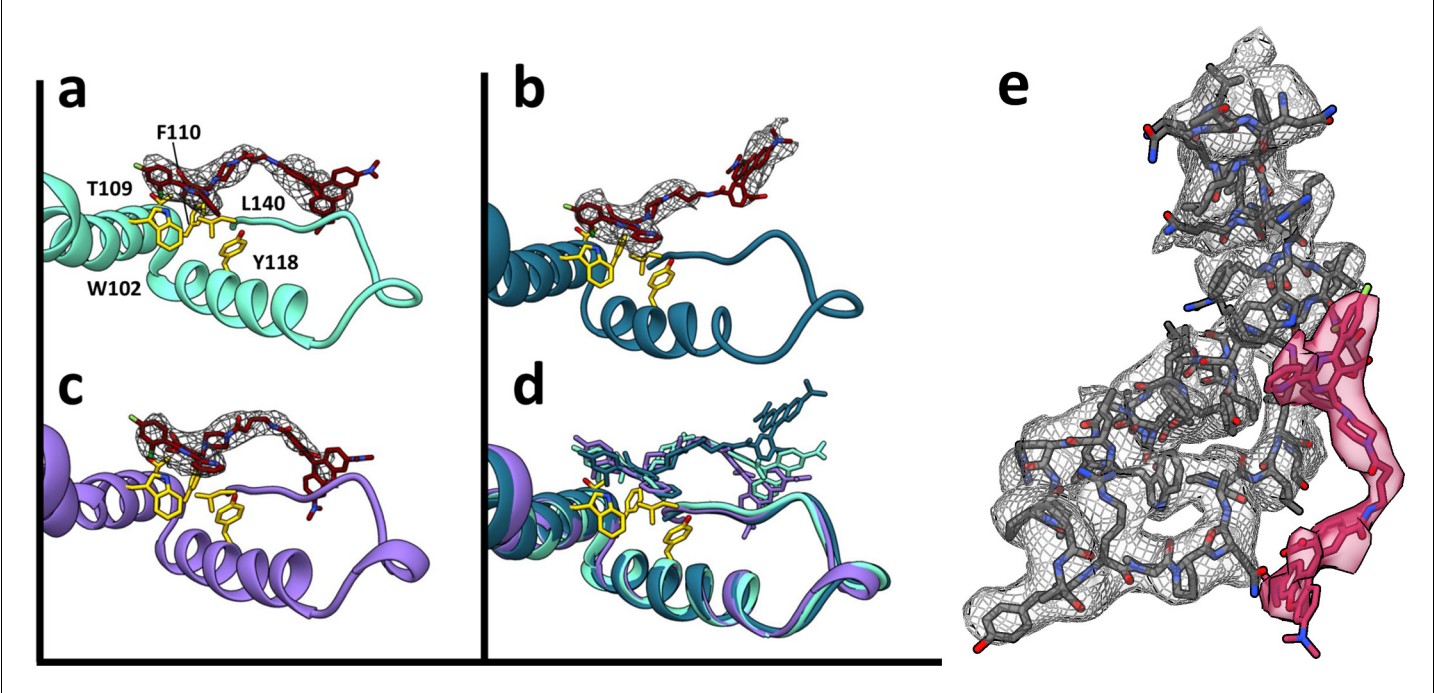

**Figure 7.** HAP-TAMRA has a very similar structure in three different quasi-equivalent environments. The HAP site consists of a pocket in one subunit that is capped by a neighboring subunit. The pockets of the T = 4 B and C subunits (panels a and b) and the T = 3 B subunit (c) each has unambiguous density for the HAP and linker moieties of HAP-TAMRA at a contour level of 3.6 σ. The B subunits of T = 3 and T = 4 show well defined density for the TAMRA moiety. The TAMRA density of the T = 4 C subunit is more ambiguous (b), suggesting multiple orientations. We show a fit consistent with available density, and note that the HAP and TAMRA regions become contiguous by 2.6σ contour. When the HAP pockets are compared, based on overlaying the respective protein monomers (d), we observe that the HAP and linker moieties have essentially the same conformation in all three sites. (e) The B site for the T = 4 structure, from a different perspective than shown in panel (a), showing density for protein and HAP-TAMRA (contoured a 3.6σ). Note that Y132, the lower left of this view, is in the peptide turn on the right side of panels a-d.

DOI: https://doi.org/10.7554/eLife.31473.010

The resolution of the TAMRA density was poorer, about 5 Å. This same pattern was also observed in the T = 3 capsid structure.

Studies with crosslinked Cp150 suggest the hypothesis that CpAM-induced structural changes and fluctuations would be concentrated at fivefold vertices and substantially greater without the constraint of the interdimer C150 disulfide. For this reason, we performed a 3D reconstruction of the heterogeneous complexes of Cp149 and HAP-TAMRA (*Figure 11a*), using data exemplified in *Figure 4a and c*. The Cp149 +HAP TAMRA reconstruction could only reach 22 Å resolution, which is a likely outcome from icosahedral averaging a wide array of structural states. The striking feature of the reconstruction is the absence of density at the fivefold. This may arise because the A subunits are extremely mobile or, more likely, that the individual A subunits are relatively static but in a variety of positions. This second hypothesis is consistent with the heterogeneity of the class averages in *Figure 4c*. This result suggests that the A subunits are the favored site of capsid failure. A comparator structure calculated to the same resolution shows well-ordered fivefold density (*Figure 11b*).

## Discussion

CpAMs, exemplified here by HAP-TAMRA, can distort capsid shape. HAP-TAMRA with crosslinked Cp150 dimers yielded capsids that retained icosahedral symmetry and were suitable for relatively high-resolution reconstruction. HAP-TAMRA with the uncrosslinked capsids of Cp149 led to asymmetric faceting, severe distortions, and broken capsids. There are advantages to examining both Cp150 and Cp149 CpAM complexes, one for structural detail and the other for biological relevance (*Wang et al., 2015*). In HBV infections, core protein exerts many of its functions while in the capsid

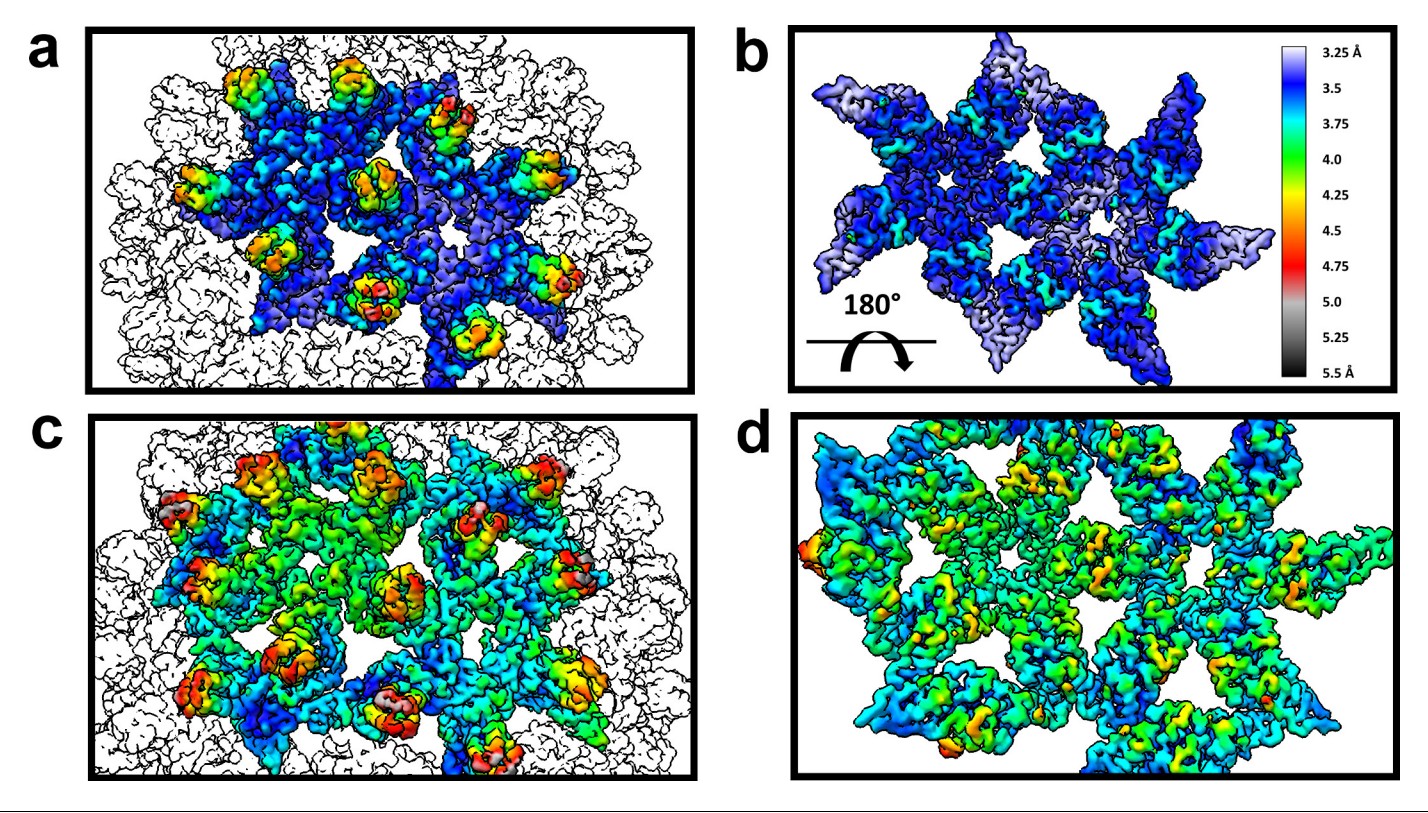

**Figure 8.** The dimer-dimer interactions, including the HAP-pocket, are structurally conserved. T = 3 (**a,b**) and T = 4 (**c,d**) capsids are color coded by resolution. Spike tips are the lowest resolution features at 4.5–5 Å resolution. The interdimer interfaces near the HAP pockets and hydrophobic cores (i.e. the dimer chassis subdomain) at the intradimer interfaces are the highest resolution regions of both capsids. Variation of resolution is analogous to differences in crystallographic B factor.

DOI: https://doi.org/10.7554/eLife.31473.011

form. Defects in the capsid structure due to CpAMs may alter exposure of signals for cellular trafficking (*Kann et al., 2007*; *Chen et al., 2016*) or dysregulate capsid dissociation (*Rabe et al., 2009*). Core protein and capsid play active roles in reverse transcription which we would expect to also be compromised in a deformed capsid (*Nassal, 2008*; *Hu and Seeger, 2015*; *Liu et al., 2015*). Core protein mutations that fill the HAP site lead to defects in both RNA packaging and transcription of the DNA plus-sense strand suggesting that CpAMs can have similar action (*Tan et al., 2015*). Though there have been reports of HAPs causing capsids to dissociate (*Stray and Zlotnick, 2006b*), the structural basis for this paradoxical effect of stabilizing Cp-Cp interaction but inducing capsid failure have, with this report, only now begun to be explored.

The T = 4 Cp150+HAP TAMRA cryo-EM structure has a larger diameter than crystallographic structures of the apo capsid (*Wynne et al., 1999a*) and CpAM-bound capsids (*Venkatakrishnan and Zlotnick, 2016*). The HAP pockets show little structural difference in quasi-equivalent sites in capsids or in crystal structures of the assembly-incompetent mutant Y132A (*Packianathan et al., 2010*; *Zhou et al., 2017*; *Klumpp et al., 2015*). This indicates that CpAM is affecting structure primarily by modulating the protein-protein interactions of the capping subunit. In addition, CpAMs can subtly modify the tertiary and quaternary structure of an individual dimer (*Venkatakrishnan and Zlotnick, 2016*). In broad terms, CpAMs flatten sixfold; during assembly this can result in non-capsid structures with extended regions of hexagonal sheet (*Stray et al., 2005*). or cylinders with a hexagonal repeat (*Liu et al., 2017*).

The flattening of quasi-sixfold, observed in molecular detail in the crosslinked Cp150 capsid, is also evident in uncrosslinked Cp149 capsids as flat regions and sharp angles. The sharp angles are likely to arise at fivefold. This is analogous to phage P22 which is assembled as a rounded

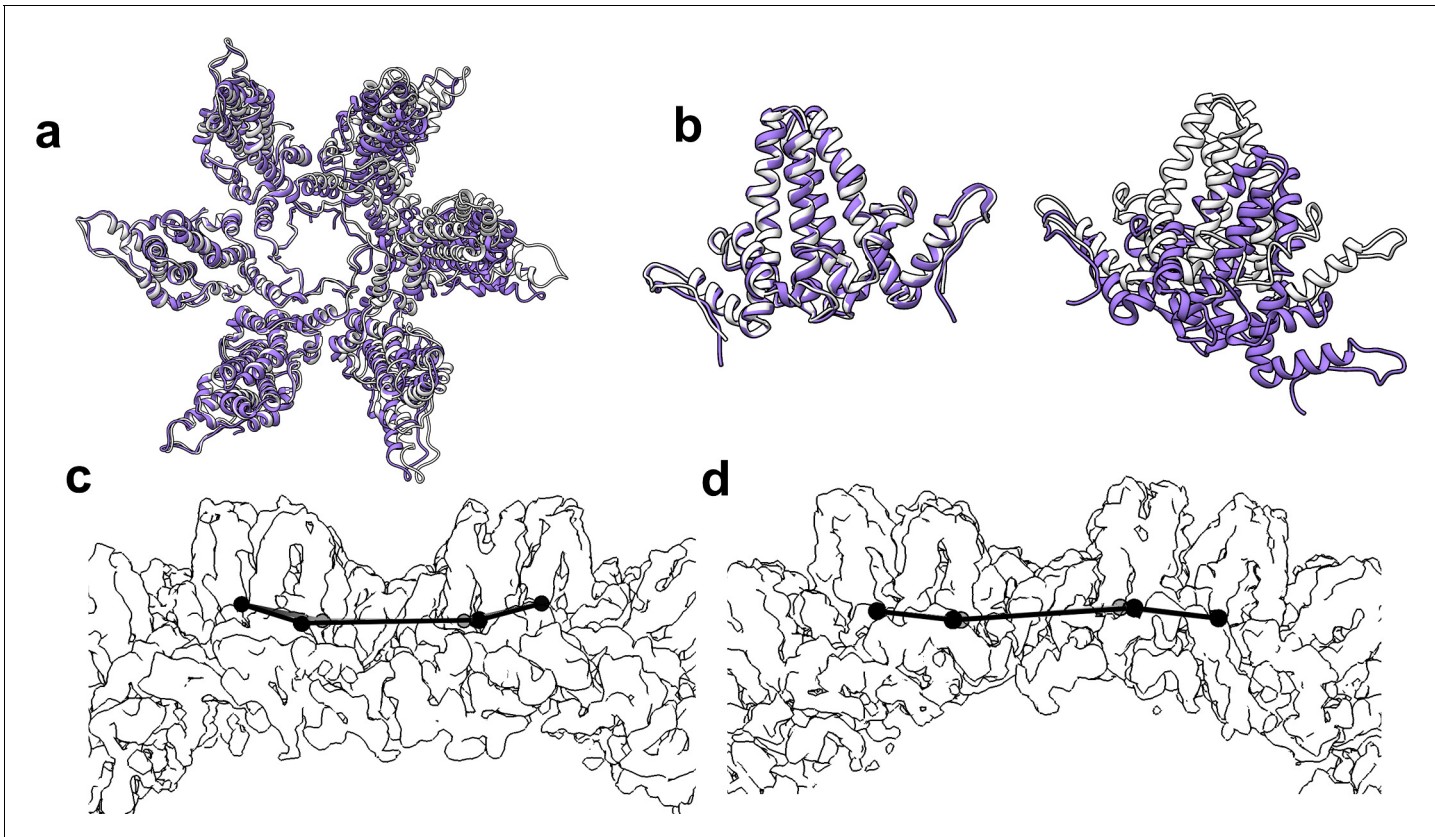

**Figure 9.** The flexibility of interdimer geometry is demonstrated by comparison of T = 3 and T = 4 quasi-sixfold vertices. Dimer–dimer interaction varies in response to quasi-equivalent interaction, T number, and the presence of CpAM. The difference in the geometry of interaction can be difficult to see. (a) Quasi-sixfold clusters of dimers from the T = 4 (white) and T = 3 (purple) structures are overlaid based on the leftmost dimer; in both cases this is an AB dimer. (b) The aligned dimers (left) are nearly identical. The dimer opposite shows a dramatic difference in 4° structure imposed by a series of relatively subtle differences. (c, d) Side views of T = 4 (c) and T = 3 quasi-sixfold show that their 4° structures are analogous to the boat and chair conformations, respectively, associated with 6-membered rings in organic chemistry (e.g. cyclohexane).

DOI: https://doi.org/10.7554/eLife.31473.012

procapsids that matures to have flattened sixfold and sharply angled fivefold (*Teschke and Parent, 2010*). In the 22 Å resolution image reconstruction (*Figure 10c*), the absence of density at fivefold could be due to highly dynamic A subunits, or to relatively static structures where the A subunits have multiple orientations. In the 2D class averages of Cp149+HAP TAMRA capsids (*Figure 4c*) most classes are asymmetrically elliptical and/or have a periphery punctuated by sharp angles, suggestive of faceting. Consider the effect of flattening a quasi-sixfold in a T = 4 HBV capsid: the A subunit in the adjacent fivefold would protrude and the C and D subunits in adjacent quasi-sixfold would be flattened. The effect on the adjacent quasi-sixfold could lead to a large flat area, resulting in a faceted or elliptical particle, or a discontinuity where the adjacent quasi-sixfold was folded. In the extreme case, such flattening could rupture a particle as observed in micrographs of Cp149 +HAP TAMRA (*Figure 4a*).

This study also has implications for the study of the physics of nanoscale materials. For a flattened quasi-sixfold to disrupt a complete capsid, retaining the flattened region must be energetically more favored than maintaining the global capsid structure. CpAMs are known to strengthen protein-protein interaction energy (*Zlotnick et al., 2015*; *Venkatakrishnan and Zlotnick, 2016*). Making the capsid surface stiffer is predicted from condensed matter theory to promote buckling transitions that lead to faceting (*Lidmar et al., 2003*; *Klug et al., 2006*). We therefore suggest that crosslinking Cp150 preserved capsid symmetry by allowing a compromise where CpAMs partially flattened surfaces but the stress was distributed globally, consistent with the mechanical principles of tensegrity structures. Tensegrity structures can also be designed to undergo structural transitions in response

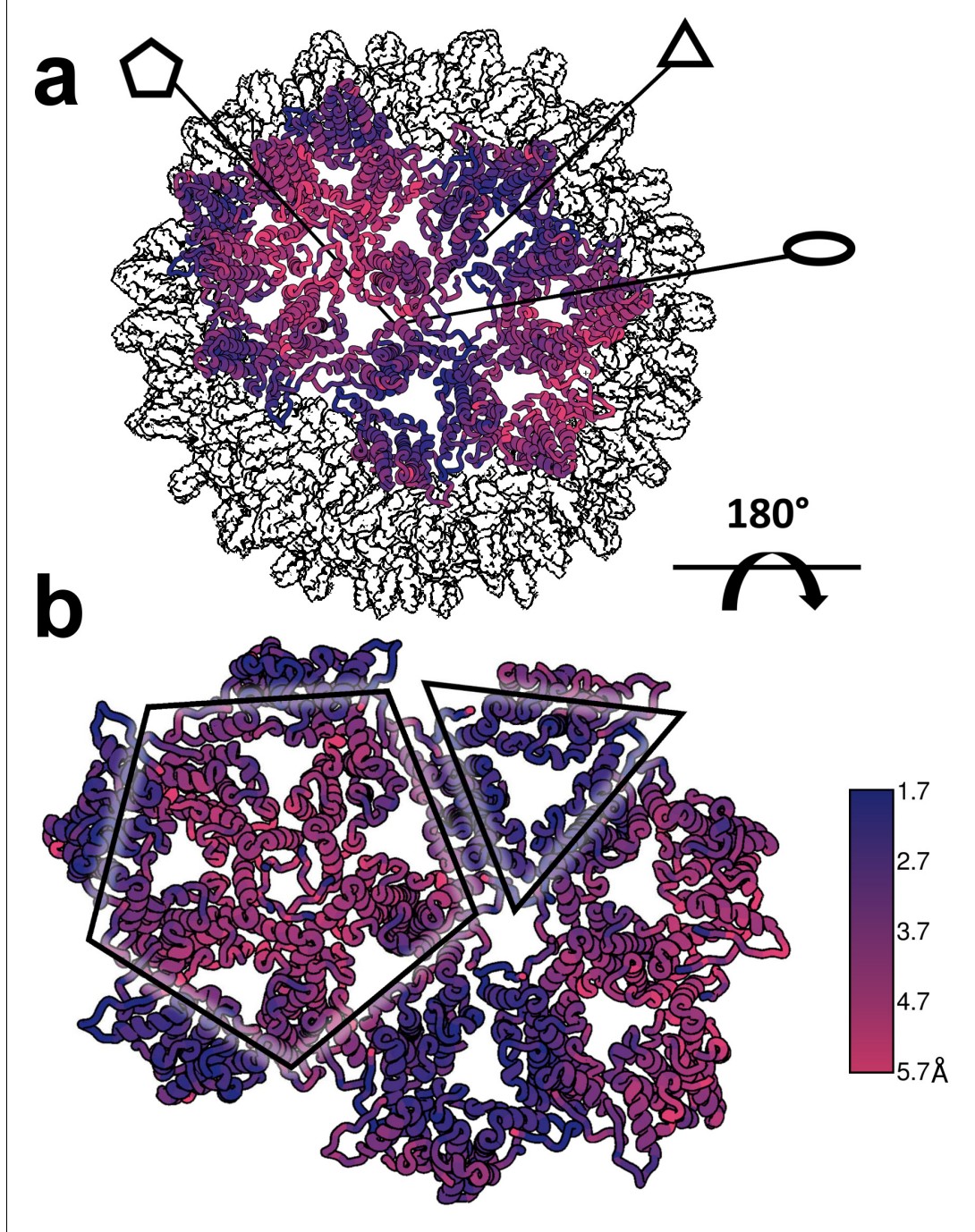

**Figure 10.** Structural defects induced by HAP-TAMRA are concentrated at icosahedral fivefold. (a, b) Differences between the HAP-TAMRA capsid and an apo capsid (1QGT) were calculated by overlaying the two capsids based on a common center. Residues are color coded based on the displacement of the α carbons. Though the HAP-TAMRA capsids are systematically larger than the apo capsids, the smallest displacement is at the base of the CD dimers surrounding icosahedral three-fold (triangles in panel b). The A subunits surrounding the fivefold vertices (pentagons in panel b) show the greatest displacement.

DOI: https://doi.org/10.7554/eLife.31473.013

to relatively small stimuli (*Domitrovic et al., 2013b*; *Skelton et al., 2001*; *Fuller, 1975*) in the same way that HBV responds to CpAM binding. Indeed, a reoccurring point in literature on the mechanical

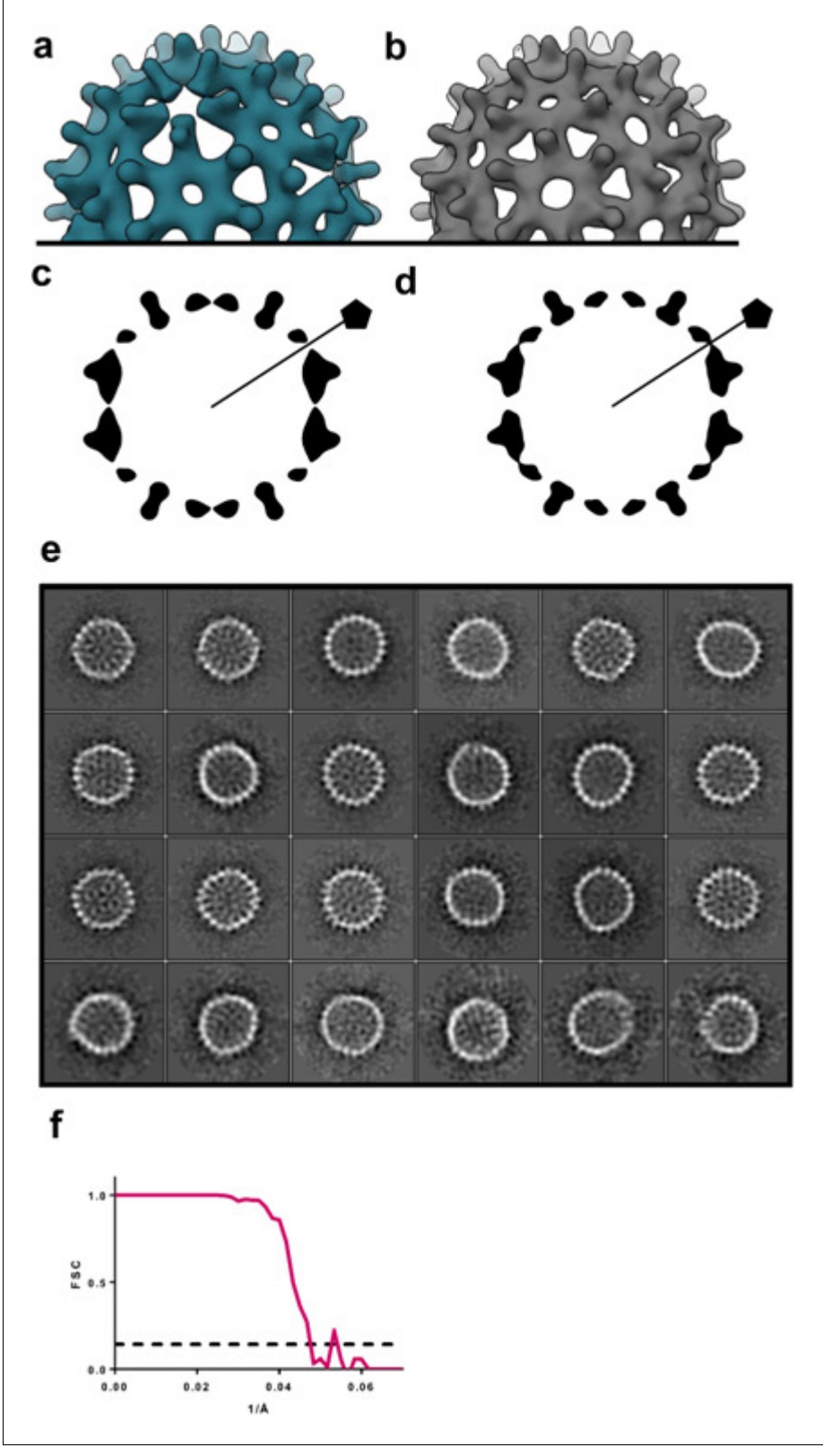

**Figure 11.** A low resolution reconstruction of Cp149+HAP TAMRA shows that defects are concentrated at fivefold vertices. (a) T = 4 reconstructions Cp149+HAP TAMRA show an absence of density on the fivefold. This suggests structural heterogeneity. Consistent with this, the reconstruction only achieved 22 Å resolution. (b) A density map based on the 1QGT molecular model down-sampled to 22 Å resolution shows the resolved features of a well-

*Figure 11 continued*
ordered fivefold. (**c,d**) Central slices through the experimental Cp149+HAP TAMRA map (**c**) and the 1QGT render
(**d**). The effect of HAP-TAMRA was to angle the dimers surrounding the fivefold axis towards twofold axes. (**e**) 2D
class averages of the particles collected for the reconstruction in panel a, sorted by descending class distribution
demonstrate faceting and irregular morphology. (**f**) FSC for reconstruction of Cp149+HAP-TAMRA.
DOI: https://doi.org/10.7554/eLife.31473.014

properties of icosahedral virus capsids, is that their stability is fundamentally related to their symmetry (*Klug et al., 2006*; *Zandi et al., 2004*; *Zandi and Reguera, 2005*).

The motivation to construct a fluorescent CpAM was that it would expand our ability to evaluate CpAM-Cp interaction in vitro and in vivo. We used changes in the absorbance profile, as well as fluorescent quenching, to observe CpAM binding to capsid. The photochemical effects we describe has been well-documented for TAMRA and other rhodamine dyes. It is a consequence of changes in electronic excitation due to proximity and dipole alignment of two or more aromatic systems (*Bergström et al., 2002*; *Ogawa et al., 2009*).

Taking advantage of HAP-TAMRA optical properties, we observed that HAP-TAMRA bound capsid tightly and slowly. Slow binding suggests that sites are only rarely open to CpAMs. The slow rate of binding is reminiscent of a proteolytic analysis of HBV capsid stability where it was observed that Cp149 capsids would undergo a slow partial unfolding transition (*Hilmer et al., 2008*). The half-life of the unfolding transition for cleavage was on the order of 100 min. In contrast, intact capsids can persist in solution for months (*Uetrecht et al., 2010*). All-atom molecular simulations of capsids indicate that HBV capsid structure is highly dynamic and is capable of departure from icosahedral symmetry (*Perilla et al., 2016*).

The multidimensional nature of a continuously deforming virus capsid (composed of 120 interconnected dimers) frustrates a rigorous treatment of binding. However, we have identified some important constraints by relating HAP-TAMRA conformation to binding cooperativity. The $\Delta$A520 signal we observe depends on formation of a $\pi$-stacked dimer (*Adachi et al., 2014*). The absorbance shift we observe (*Figure 3f*) indicates that all of the bound HAP-TAMRA is involved in $\pi$-stacked dimers, even when there is on average approximately a single HAP-TAMRA per quasi-sixfold vertex. This would not occur if HAP-TAMRA were randomly distributed over a capsid. In a random distribution at the lowest concentration we tested only about 1/3 of the probe would be in quasi-sixfold vertices with two or more bound molecules. Furthermore, the structure of the T = 4 capsid suggests interaction only between probes in the B and C' or C and B' sites (*Figure 5b*). Of the six possible pairwise interactions for a T = 4 hexamer with two sites filled (BB', BC', BC, CB', CC, B'C'), only one third is expected to yield a $\Delta$A520 signal. Thus, at the lowest HAP-TAMRA:dimer ratio tested, random binding predicts 1/3*1/3 = 1/9 or 11% of the maximum possible $\Delta$A520. Nonetheless, we observe approximately 100% of the possible signal. This indicates a high degree of cooperativity of binding, at least at a local level.

Binding kinetics also suggests some degree of cooperativity. We would expect to see sigmoidal kinetics if sites filled randomly. Thus, once a B site is filled there must be a preference for filing an adjacent C' with a much faster rate constant (or for a C site a preference for an adjacent B'). In a slightly more complicated model, B and C may have different rate constants for the initial binding event. This second model could fit to our data with two first order curves of equal amplitude (*Figure 3b*). However, this local structural change model does not account for the global structural change we observe. Furthermore, contrary to Occam's razor, a simple fit may obscure complex, multi-phase kinetics. If we consider that flattening one quasi-sixfold will affect the structure of the neighboring quasi-sixfold, we can see how capsid asymmetry is induced. Asymmetric distortion of quasi-sixfold will affect binding site structures and therefore their kinetics of binding and affinities. There may be a continuum of binding sites from flattened quasi-sixfold, as might be found in a large aberrant structure or the flat areas of an oblate ellipse (i.e. large radius of curvature), to bent quasi-sixfold that might be found where there is a small radius of curvature. The result of this continuum of sites will not fit a single first order rate law, but may be approximated by as the sum of two or more (*Zlotnick et al., 1994*).

In a broad sense, our results reaffirm the view that virus capsids are not inert containers. It is likely that some of the most important biological functions of icosahedral capsids may occur when they

become non-icosahedral. We have demonstrated that non-icosahedral states of HBV capsids can be induced using CpAMs. Indeed, CpAMs can deform and even disrupt intact capsids, which suggest modes of action beyond the established mechanisms of assembly activation and misdirection. A molecule which simultaneously affects two or more stages of the lifecycle would reduce the impact of escape mutations, and increases the likelihood of clearing an infection.

## Materials and methods

### HBV sample preparation

The Hepatitis B subtype *adyw* core protein assembly domain and the Cp150 mutant, in which three native cysteine are mutated to alanine and an additional C-terminal cysteine appended, were expressed in *E. coli* and purified by size exclusion as previously described (*Zlotnick et al., 2007*). Cp149 and Cp150 capsids were prepared for preliminary Cryo-EM from purified dimer by initiating assembly at 10 µM dimer concentration in 300 mM NaCl, 50 mM HEPES, pH 7.5 and allowing the reaction to proceed overnight. Residual un-assembled dimer was removed by purifying the fresh capsids via size exclusion. Purified capsids were then incubated with HAP-TAMRA at a molar excess overnight. For cryo-EM, capsids were then concentrated to >10 mg/ml.

### Synthesis of HAP-TAMRA

Synthesis of HAP13 (*Bourne et al., 2008*) and HAP-TAMRA are described in the patent literature (*Zlotnick et al., 2014*). Analysis of the product by HPLC at the 280 nm and 555 nm wavelengths showed a product with ~99% purity; ESI-TOF MS: calculated for $C_{51}H_{50}ClFN_8O_7$, $m/z$ 940.35; $M^+$ + H, $m/z$ 941.36; Found, $m/z$ 941.3; calculated for $M^+$ + Na, $m/z$ 963.4; Found, $m/z$ 963.3.

### Negative stain electron microscopy

Samples from *Figure 2a,b* represent Cp149 assembly reactions at 50 mM HEPES, 300 mM NaCl, and 10 µM dimer. The reaction in *Figure 2b* was initiated in the presence of 40 µM HAP-TAMRA. Both samples were adsorbed to the surface of EMS carbon film 300 mesh coper grids, washed with water, and stained with 2% uranyl acetate. Samples were imaged with a JEOL 3200-FS electron microscope operated at 300kV.

### Cryo-electron microscopy and image processing

Samples for Cryo-EM were applied to a glow-discharged Quantifoil holey-carbon grids (R2/2). The grids were blotted with filter paper for 4 s before automated plunging into liquid ethane using an FEI vitrobot. The Cp149+HAP TAMRA data from *Figure 11* and *Figure 4* panel a were imaged using a JEOL 3200-FS electron microscope operated at 300kV with an in-column energy filter. Images were recorded at a nominal magnification of 80,000X, with a pixel size of 1.5 Å, maintaining less than 35e⁻ Å² dose, recorded on a Gatan Ultrascan 4000 4k × 4 k CCD detector. The 2D class averages shown in *Figure 4* panels b, c, and d were collected on the same JEOL 3200-FS microscope, but with a Direct Electron DE-12 detector, with an image sampling rate of 1.01 Å/pixel. For the high-resolution 3D structure determination, first presented in *Figure 5*, samples were imaged on a FEI Titan Krios operated at 300 kV at a nominal magnification of 22,500. Images were recorded on a Gatan K2 Summit detector operating in 'super-resolution' mode, resulting in a pixel size of 0.65 Å with a dose of ~33 e⁻ Å². Each exposure was 8 s long, and was collected as 35 individual frames. The data collection process was semi-automated using the Leginon system.(*Suloway et al., 2005*) The final maps are deposited in the EMDB database as 7295 for the T = 3 map and 7294 for the T = 4 map (*Table 1*).

Cryo-EM classification and reconstruction was implemented using standard protocols of the EMAN2, Motioncorr2, and Relion software programs (*Kimanius et al., 2016*; *Zheng et al., 2016*; *Tang et al., 2007*). Upon convergence of the 3D structures, each map was sharpened by applying a negative B-factor which was obtained using the Guinier fitting procedure implemented in Relion (*Rosenthal and Henderson, 2003*). To assess the local variability in the quality of the structure a local resolution analysis was employed, also implemented in Relion. The local resolution procedure determined local FSC with a sampling window of 10 Å, using the same negative B-factor obtained at the end of refinement.

## Model determination

The crystal structures of both apo capsid (1qgt)(*Wynne et al., 1999b*) and HAP18 bound capsid (5d7y)(*Venkatakrishnan et al., 2016*) were used as starting points for flexible model refinement imposing icosahedral non-crystallographic symmetry constraints, and using the PHENIX and eLBOW software programs (*Afonine et al., 2012*; *Moriarty et al., 2009*; *Emsley et al., 2010*). The model validation statistics we report were obtained from the final output of the Phenix real space refinement tool. To refine the pixel size of the map, we fit the model of the asymmetric unit into density and measured cross-correlation across pixel sizes. The optimal pixel size was determined to be 1.285 Å, compared to 1.30 Å as determined by the microscope calibration. The final maps and models reflect this change. Structural comparisons and figure creation were carried out in UCSF Chimera. The final structure coordinates are deposited in the protein data bank as 6BVN for the T = 3 structure and 6BVF for the T = 4 structure.

## Detection of binding in solution

The final buffer conditions for all binding assays were 300 mM NaCl, 20 mM Tris, 1% DMSO, and pH 7.5 with varying protein and HAP-TAMRA. Size exclusion chromatography assays were performed using a superose 6 30/10 column plumbed into a Shimadzu HPLC equipped with a diode array detector. The HAP-TAMRA absorbance eluting in the capsid fractions was attributed to the capsid bound population. All HAP-TAMRA absorbance eluting later was attributed to free ligand. It is important to note that because the HAP-TAMRA probe is synthesized as a racemic mixture, and because only a single enantiomer is known to bind the HAP pocket, we expected to see at least half of the input HAP-TAMRA eluting as free ligand. When plotting the increase in 520 nm signal in the capsid fraction (*Figure 2d*), the A520 at 7.5 ml was used, corresponding to the pre-established center of the capsid elution volume.

Absorbance and fluorescence was measured in a 96-well plate using a 200 µl sample volume by a Biotek Synergy H1 plate reader. Measurements based on fluorescence used an excitation wavelength of 520 nm and monitored emission at 580 nm. For plotting the change in 520/555 nm absorbance ratio, we used the maximum value of the capsid peak at each wavelength. Kinetic assays based on absorbance were sampled in 1 min intervals, where each data point is the value of 520 nm absorbance. To account for the varying presence of excess free probe in the titration time course (*Figure 3b*), the signal is presented as $\Delta$520 nm absorbance compared to a probe-only reference. We defined this as $\Delta A520 = A520_{raw} - A520_{inert} - A520_{binding(t=0)}$ where $A520_{inert}$ accounts for the absorbance of dye that does not participate in binding (because it was either the inactive enantiomer or present in superstoichiometric quantities), and $A520_{binding(t=0)}$ is the initial absorbance of the fraction of HAP-TAMRA that will participate in binding. This analysis assumes linear binding until saturation, based on the results in *Figure 2d*, but could underestimate the signal if binding were weaker.

## Acknowledgements

We would like to acknowledge critical discussions with Dr. Shefah Qazi regarding TAMRA photophysics. The IU Electron Microscopy Center (IUEMC), notably Dr. David Morgan, and the Purdue cryo-EM facility at Purdue University, notably Drs. Thomas Klose and Valerie Bowman, provided invaluable support. HHMI and Assembly Biosciences contributed to upgrading the facilities at IUEMC. This work was supported by NIH R01-AI067417 and a supported research agreement from Assembly Biosciences to AZ.

## Additional information

### Competing interests

Christopher John Schlicksup: AZ has an interest in Assembly Biosciences, a company that is pursuing antivirals directed against HBV. Balasubramanian Venkatakrishnan: SF is an employee of Assembly Biosciences, a company that is pursuing antivirals directed against HBV. Michael VanNieuwenhze:

WW is an employee of Assembly Biosciences, a company that is pursuing antivirals directed against HBV. The other authors declare that no competing interests exist.

## Funding

| Funder | Grant reference number | Author |
|---|---|---|
| National Institute of Allergy and Infectious Diseases | R01-AI067417 | Adam Zlotnick |
| Assembly Biosciences | | Adam Zlotnick |

The funders had no role in study design, data collection and interpretation, or the decision to submit the work for publication.

## Author contributions

Christopher John Schlicksup, Conceptualization, Resources, Formal analysis, Supervision, Funding acquisition, Writing—original draft, Project administration, Writing—review and editing; Joseph Che-Yen Wang, Conceptualization, Formal analysis, Validation, Investigation, Visualization, Methodology, Writing—original draft, Writing—review and editing; Samson Francis, Conceptualization, Formal analysis, Supervision, Investigation, Visualization, Writing—review and editing; Balasubramanian Venkatakrishnan, Resources, Investigation, Methodology, Writing—original draft; William W Turner, Formal analysis, Visualization, Methodology; Michael VanNieuwenhze, Conceptualization, Resources, Methodology; Adam Zlotnick, Resources, Funding acquisition

## Author ORCIDs

Adam Zlotnick (iD) http://orcid.org/0000-0001-9945-6267

## Decision letter and Author response

Decision letter https://doi.org/10.7554/eLife.31473.027
Author response https://doi.org/10.7554/eLife.31473.028

# Additional files

## Supplementary files

• Transparent reporting form
DOI: https://doi.org/10.7554/eLife.31473.017

## Major datasets

The following datasets were generated:

| Author(s) | Year | Dataset title | Dataset URL | Database, license, and accessibility information |
|---|---|---|---|---|
| Christopher John Schlicksup, Joseph Che-Yen Wang, Adam Zlotnick | 2017 | Cryo-EM Structure of Hepatitis B virus T=4 capsid in complex with the fluorescent allosteric modulator HAP-TAMRA | http://www.rcsb.org/pdb/search/structid-Search.do?structureId=6BVF | Publicly available at the RCSB Protein Data Bank (accession no. 6BVF) |
| Christopher John Schlicksup, Joseph Che-Yen Wang, Adam Zlotnick | 2017 | Cryo-EM Structure of Hepatitis B virus T=3 capsid in complex with the fluorescent allosteric modulator HAP-TAMRA | http://www.rcsb.org/pdb/search/structid-Search.do?structureId=6BVN | Publicly available at the RCSB Protein Data Bank (accession no. 6BVN) |
| Christopher John Schlicksup, Joseph Che-Yen Wang, Adam Zlotnick | 2017 | Cryo-EM Structure of Hepatitis B virus T=4 capsid in complex with the fluorescent allosteric modulator HAP-TAMRA | http://www.ebi.ac.uk/pdbe/entry/emdb/EMD-7294 | Publicly available at the Electron Microscopy Data Bank (accession no. EMD-7294) |
| Christopher John Schlicksup, Joseph Che-Yen Wang, Adam Zlotnick | 2017 | Cryo-EM Structure of Hepatitis B virus T=3 capsid in complex with the fluorescent allosteric modulator HAP-TAMRA | http://www.ebi.ac.uk/pdbe/entry/emdb/EMD-7295 | Publicly available at the Electron Microscopy Data Bank (accession no. EMD-7295) |

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
