## [Decision Letter]

Thank you for submitting your article "Hepatitis B Virus Core Protein Allosteric Modulators Can Distort and Disrupt Intact Capsids" for consideration by *eLife*. Your article has been reviewed by three peer reviewers, one of whom is a member of our Board of Reviewing Editors, and the evaluation has been overseen by Arup Chakraborty as the Senior Editor. The reviewers have opted to remain anonymous.

The reviewers have discussed the reviews with one another and the Reviewing Editor has drafted this decision to help you prepare a revised submission. Detailed comments are below.

Summary:

This manuscript describes the effect of molecules that have previously been shown to hyperstabilze assemblages of hepatitis B virus (HBV) capsids. A fluorescent version was synthesized and used to show binding to a truncated version of assembled capsid protein by both co-purification and fluorescence quenching. Distortions were observed in cryo-EM that were sufficiently great that the authors reconfigured the capsid protein used for assembly to stabilize mobile residues at the C terminus (Figure 4). Now, the distortions could be better visualized, and the structures of particle sets solved, by cryoEM. This is a very important piece of work that shows distorting of an highly complex oligomer with a small inhibitor molecule. The structural comparisons in Figure 7–Figure 11 are great, and show that the distortions are at the five-fold axis.

In this manuscript Schlicksup et al. examine the effect of the binding of a fluorescent derivative of HAP (HAP-TAMRA) on preformed hepatitis B virus capsids. The main findings are that binding of the molecule distorts the capsid and that the binding of HAP-TAMRA is slow.

In this manuscript by Schlicksup et al., the authors used cryo-EM and absorbance-based binding assays to determine the structural impact of fluorophore-labeled heteroaryldihydropyrimidines (HAP) binding to the Hepatitis B Virus core protein. The major findings illustrate: (1) the capacity of these molecules to distort capsid shape with non-crosslinked material, (2) the high-resolution cryo-EM structures of cross-linked complexes associated with the HAP-TAMRA to define the locations and influence on capsid structure, and (3) to illustrate the inherent flexibility that correlates to functionality of virus capsids and their assembly process.

Essential revisions:

Major comments on Figures

1) Figure 2:

a) Should be Figure 1. Synthesis of fluorescent TAMRA derivative is not mentioned in Results and should be discussed in the text. It pertains to a sentence on the first line of the Results, and to the hyper-stabilization of capsid structures in what is currently Figure 1. Therefore, the synthesis should no doubt be Figure 1, perhaps combined with the EM stabilization figure.b) Does "chart 2" in the Materials and methods refer to Figure 2?c) The X-axis of Figure 1 is labeled on the right rather than the left. The fonts are all different.d) Figure 1 which uses the shift in absorbance of HAP-TAMRA to quantify binding presented before the shift in absorbance is validated (Figure 3).

2) Figure 1: Should be Figure 2. It would be good to see a micrograph of unbound capsids in addition to the distorted ones so that the non-specialist can appreciate the distortion observed. The authors assert that HAP-TAMRA accelerates assembly, stabilizes the assembled products, and result in the formation of aberrant particles. For example, the legend to Figure 1 say "HAP-TAMRA binding drives core protein assembly " and yet two of the four panels show binding to preformed cores. It would be ideal to show data for all three claims but a minimal approach would be that the experiments (assembly kinetics and stability) be described in the Materials and methods.

3) Figure 3 is described in terms of (A), (B) and (C) but these are not marked. Several other comments include:a) Figure 3 and Figure 3 should be mentioned in the text.b) In Figure 3, the legend "the weighted sum of free and bound TAMRA" is difficult to understand. Of absorbance at 550? Wouldn't that require knowing the extinction coefficient for both species? At what time point were the samples taken?c) In Figure 3, what is the inset and what were the conditions used for the time course?d) Figure 3 is very nice and follows directly on Figure 3 (which determined the conditions), probably should follow Figure 3 in the presentation.e) The analysis of the binding kinetics needs improvement. A trace is shown with a half-time of approximately 10 min and the statement is made that the kinetics are complex. A better analysis would require following the binding kinetics at multiple HAP-TAMRA concentration, perhaps showing a change in rate-determining step when the putative open motion became rate limiting. The claim that the kinetics are "complex" is not meaningful without an explanation for the claim. Were fits attempted? What sort of fits, and with what results?

4a) In Figure 4, it would be better to see images of the T=3 structures as well, as we are going to see their structures in the next figures.b) The conclusion in the Abstract that the data demonstrate that capsid are tensegrity structures (as originally suggested by Caspar) is not discussed in the manuscript.c) A scale bar is needed.

5) In Figure 5, (A) and (C) should be (A) and (B) and similarly for (B) and (D) should be (C) and (D). In (A) the colors are not described accurately. The nomenclature "AB dimer" and "CD dimer" should be better explained for non-specialized readers. In (A) it would help if the quasi-sixfold axis were named as such before directing our attention to the five-fold axis.

6) The table of reconstruction metrics is presented as a Figure instead of a Table,

7) In Figure 8 legend, the TAMRA density is ambiguous in the C subunit of the T=4 structure, it is modeled like in the B subunits. Then it is concluded that the overlays are similar.. What happened to the C subunit of the T=3 structure? The quasi-symmetry is very interesting and should be better discussed for the non-specialist.

8) The cryo-EM reconstructions and the supplemental movies are very nice.

Major Comments on Text

The main criticism is that the text does not track well with the figures, some of the figures are mis-labeled and the text is terse, making what is an elegant paper surprisingly hard to read. Careful editing is needed, tracking the text to the figures and tightening the logical flow.

---

## [Author Response]

In this manuscript Schlicksup et al. examine the effect of the binding of a fluorescent derivative of HAP (HAP-TAMRA) on preformed hepatitis B virus capsids. The main findings are that binding of the molecule distorts the capsid and that the binding of HAP-TAMRA is slow.In this manuscript by Schlicksup et al., the authors used cryo-EM and absorbance-based binding assays to determine the structural impact of fluorophore-labeled heteroaryldihydropyrimidines (HAP) binding to the Hepatitis B Virus core protein. The major findings illustrate: (1) the capacity of these molecules to distort capsid shape with non-crosslinked material, (2) the high-resolution cryo-EM structures of cross-linked complexes associated with the HAP-TAMRA to define the locations and influence on capsid structure, and (3) to illustrate the inherent flexibility that correlates to functionality of virus capsids and their assembly process.

In response to the reviewers’ suggestions regarding a more detailed analysis of HAP-TAMRA interaction with capsids we have observed substantive evidence of cooperative binding (Figure 3). This is an important point for two reasons. First it provides a biophysical mechanism for transducing binding of the small molecule to distortion of the capsid; binding events induce local structural changes that lead to stronger subsequent binding. Secondly, we show that the rate of binding is independent of HAP-TAMRA concentration indicating that the rate limiting step of binding depends on “opening” binding sites. A reaction that we suggest depends on capsid conformational change. Taken together, these data suggest a basis for progressive distortion of capsids. Most of these results are shown in Figure 3.

Essential revisions:Major comments on Figures1) Figure 2:a) Should be Figure 1. Synthesis of fluorescent TAMRA derivative is not mentioned in Results and should be discussed in the text. It pertains to a sentence on the first line of the Results, and to the hyper-stabilization of capsid structures in what is currently Figure 1. Therefore, the synthesis should no doubt be Figure 1, perhaps combined with the EM stabilization figure.

The schematic description of HAP-TAMRA synthesis is moved to Figure 1. We chose to keep data in a separate figure

b) Does "chart 2" in the Materials and methods refer to Figure 2?

This summary of HPLC data is shown graphically in Figure 2.

c) The X-axis of Figure 1 is labeled on the right rather than the left. The fonts are all different.

Fixed.

d) Figure 1 which uses the shift in absorbance of HAP-TAMRA to quantify binding presented before the shift in absorbance is validated (Figure 3).

Figure 1 (now 2C) shows spectra of HPLC peaks (capsid-bound HAP-TAMRA and free HAP-TAMRA) determined using a diode array detector. These spectra are the basis for the analysis of binding shown in Figure 2. The absorbance spectrum (Figure 3) demonstrates the effect can be measured, even without HPLC separation.

2) Figure 1: Should be Figure 2. It would be good to see a micrograph of unbound capsids in addition to the distorted ones so that the non-specialist can appreciate the distortion observed. The authors assert that HAP-TAMRA accelerates assembly, stabilizes the assembled products, and result in the formation of aberrant particles. For example, the legend to Figure 1 say "HAP-TAMRA binding drives core protein assembly " and yet two of the four panels show binding to preformed cores. It would be ideal to show data for all three claims but a minimal approach would be that the experiments (assembly kinetics and stability) be described in the Materials and methods.

We have added a micrograph of unbound capsids. Removed the claims about acceleration and stability from results.

3) Figure 3 is described in terms of (A), (B) and (C) but these are not marked.

Figure 3 is extensively revised to include new data and analyses. Panel (A) still shows the shift I the TAMRA absorbance spectrum as a function of association with capsid. In panel (B) we show the kinetics of binding at multiple HAP-TAMRA concentrations. In panel (C) we show that binding kinetics are independent of HAP-TAMRA concentration. The data in panels (B) and (C) suggest a simple binding model that is described in text. In panel (D) we examine the endpoint binding signal which indicate that binding is high affinity relative to the concentrations used in the experiment and the stoichiometry of HAP-TAM for core protein is one drug per dimer, a result consistent with the structure shown in the following studies. In panel (E) we observe that bound drug always has the absorbance shift related to stacked TAMRA moieties, even where the ratio of HAP-TAMRA to core protein is less than one HAP-TAMRA per sixfold vertex. In discussion, we point out that this indicates that HAP-TAMRA must bind with a high degree of cooperativity so that molecules are in adjacent positions. This cooperativity of binding especially in light of the distorted particles observed in cryo-EM class averages (Figure 4) implies a cooperative and progressive structural transition. Such a progressive transition is consistent with tensegrity, where stress is distributed across a structure. The progressive structural transition model also suggests a Koshland-Nemethy-Filmer model of cooperativity.

These extensive changes work in the more specific comments, below.

Several other comments include:a) Figure 3 and Figure 3 should be mentioned in the text.

Fixed.

b) In Figure 3, the legend "the weighted sum of free and bound TAMRA" is difficult to understand. Of absorbance at 550? Wouldn't that require knowing the extinction coefficient for both species? At what time point were the samples taken?

We have reworded the text and figure legend.

c) In Figure 3, what is the inset and what were the conditions used for the time course?d) Figure 3 is very nice and follows directly on Figure 3 (which determined the conditions), probably should follow Figure 3 in the presentation.

Text is completely revised.

e) The analysis of the binding kinetics needs improvement. A trace is shown with a half-time of approximately 10 min and the statement is made that the kinetics are complex. A better analysis would require following the binding kinetics at multiple HAP-TAMRA concentration, perhaps showing a change in rate-determining step when the putative open motion became rate limiting. The claim that the kinetics are "complex" is not meaningful without an explanation for the claim. Were fits attempted? What sort of fits, and with what results?

As outlined above, the kinetics have been described much more thoroughly with more data and a more complete analysis.

4a) In Figure 4, it would be better to see images of the T=3 structures as well, as we are going to see their structures in the next figures.

Class averages for T=3 and T=4 particles, both from the same micrographs used in the high resolution image reconstructions, are included in Figure 6.

b) The conclusion in the Abstract that the data demonstrate that capsid are tensegrity structures (as originally suggested by Caspar) is not discussed in the manuscript.

A brief tensegrity discussion is now included in text.

c) A scale bar is needed.

Done.

5) In Figure 5, (A) and (C) should be (A) and (B) and similarly for (B) and (D) should be (C) and (D). In (A) the colors are not described accurately. The nomenclature "AB dimer" and "CD dimer" should be better explained for non-specialized readers. In (A) it would help if the quasi-sixfold axis were named as such before directing our attention to the five-fold axis.

Terminology is now better illustrated and described in figures and in text. We followed the reviewer’s suggestion on relabeling.

6) The table of reconstruction metrics is presented as a Figure instead of a Table,

We separated the FSC curves (Figure 6) and table (Table 1).

7) In Figure 8 legend, the TAMRA density is ambiguous in the C subunit of the T=4 structure, it is modeled like in the B subunits. Then it is concluded that the overlays are similar.. What happened to the C subunit of the T=3 structure? The quasi-symmetry is very interesting and should be better discussed for the non-specialist.

We have updated the figure for clarity. In the T=4 C subunit, density for the HAP component of HAP-TAMRA is well-defined. Density for the linker and TAMRA moieties is ambiguous. However, it is clear that the linker and TAMRA moieties from the T=4 C site follow a different trajectory than those of the T=4 B site and the T=3 B site. The overlays display this difference, which is now also specified in text.

In the T=3 capsid, HAP-TAMRA is clear seen in the B subunit but is absent in the C-subunit. We now specify that the HAP pocket of the T=3 C subunit resembles the similarly empty pocket in the T=4 D subunit.

8) The cryo-EM reconstructions and the supplemental movies are very nice.Major Comments on TextThe main criticism is that the text does not track well with the figures, some of the figures are mis-labeled and the text is terse, making what is an elegant paper surprisingly hard to read. Careful editing is needed, tracking the text to the figures and tightening the logical flow.

We have added text and re-ordered figures to correct this disruption.